# Validation of the Chinese Version of the Centrality of Religiosity Scale (CRS): Teacher Perspectives

**John Chi-Kin Lee [1,2,*] and Xiaoxue Kuang [3]**

[1] Department of Curriculum and Instruction, The Education University of Hong Kong, New Territories, Hong Kong, China

[2] Director of Centre for Religious and Spirituality Education, The Education University of Hong Kong, New Territories, Hong Kong, China

[3] School of Education (Normal School) of Dongguan University of Technology, Dongguan University of Technology, Dongguan 523000, China; 2017057@dgut.edu.cn

\* Correspondence: jcklee@eduhk.hk

**Abstract:** This study applied the Centrality of Religiosity Scale (CRS) to the context of Hong Kong as a part of China with the focus on a specific target group of teachers in primary and secondary schools. For the validation of the scale in the Hong Kong context, the version of CRSi-20 was tested with data collected from local teachers (N = 671). For the validation of the scale, six versions were tested (CRSi-20, CRS-15, CRSi-14, CRS-10, CRSi-7, and CRS-5). Confirmatory Factor Analysis demonstrated that the single-factor solution of five items (CRS-5) had better fit indices than the seven-item version (CRSi-7), which, in turn, was better than CRS-15 with a five-factor solution (Intellect, Ideology, Private Practice, Public Practice, and Religious Experience). The other three versions encountered a problem with high correlations between factors. Multiple-indicators multiple-causes (MIMIC) analysis was used to test the effect of covariates on the established factor structure for CRS-5, CRSi-7, and CRS-15. The results indicated that gender and religious belief are significant predictors of the centrality of religiosity scores for CRS-5, CRSi-7, and CRS-15. In addition, age was a positive predictor for public practice, and teachers' education level was positively related to private practice for CRS-15. Implications regarding understanding for the existing literature are discussed.

**Keywords:** centrality of religiosity scale; validation; religion; Hong Kong

## 1. Introduction

### 1.1. Introduction, Theoretical Underpinning, and Psychometric Properties of the CRS

Religiosity is a complex concept with its multiple definitions from cross-disciplinary perspectives. It encompasses aspects of motivational, emotional (feelings), and behavioral characteristics (participation in different religious activities) as well as beliefs and commitments (Abbasi et al. 2019, p. 319; Hackney and Sanders 2003). From a psychological perspective, religiosity tends to be associated with devotion, holiness, and piousness (Groome 1998), and religious education focuses on students' understanding of religious viewpoints (Clayton and Stevens 2018). Sociologists would emphasize church membership and attendance and doctrinal knowledge (Cardwell 1980; Holdcroft 2006, p. 89). Different dimensions of religiosity have been defined by various scholars. From the early 1900s, sociologists and psychologists have been in agreement about the multiple dimensions of religiosity (Pearce et al. 2017). Five dimensions are identified by Glock and Stark (1965): Experiential, ritualistic, ideological, intellectual, and consequential. Four dimensions of religiosity are proposed by Fukuyama (1961): Cognitive (intellectual, knowledge), cultic (religious practices, ritual behavior), creedal (personal religious belief),

and devotional (experiential). Multidimensional measures of religiosity are also proposed (Park 2002). There are specific measures of religiosity for some religious groups. Aziz and Rehman (1996), for example, studied Muslims' constructs of religious effect, doctrine, and faith, while Wilkes et al. (1986) measured Christians' dimensions of self-perceived religiousness, the significance of religious ethics, church attendance, and assurance in religious norms or confidence in religious values (Khan 2014, pp. 69–70).

Since many measures of religiosity have been used for psychology and have been applied in organizational research (King and Crowther 2004), this paper focuses on the Centrality of Religiosity (CRS) scale, which is also widely used internationally.

The Centrality of Religiosity (CRS) scale is an instrument to measure the meaning of religious significance to the personality and was developed by Stefan Huber (Huber and Huber 2012, p. 711). It has gained worldwide attention and has been incorporated in more than 100 cross-disciplinary research works and in more than 20 countries in the global Religion Monitor. The theoretical basis of the CRS is partly based on the standpoints of Allport and Ross (1967, p. 434) with extrinsic and intrinsic religiosity, Glock's notion of distinction expressions of religious life (Zwingmann et al. 2011; Gheorghe 2019, p. 1; Glock 1962), and Kelly's (1955) views of personality psychology. The CRS refers to the "the total of religious life" and it aims at "transreligious generalization of the measure" (Huber and Huber 2012, p. 711). In addition, it encompasses the five sociologically defined dimensions that concern an individual's religious identity and cost–reward analysis of religious life (p. 713). For the personal construct theory, Kelly (2003, pp. 9–14) explained that it has eleven corollaries (adapted for religiosity in the case of CRS): (1) Construction: Personal anticipation of religious events by perceiving the similarities with some past events (construing their replications); (2) Individuality: Personal difference in construction of religious events; (3) Organization: Evolution of the construction system covering connection between the five constructs related to religiosity in the CRS; (4) Dichotomy: Personal construction system of religiosity covering a limited number of dichotomous constructs for different groups; (5) Choice: Choosing alternatives for religious activities and experiences in a dichotomous construct; (6) Range: Personal anticipation of a finite selection of religious activities; (7) Experience: Variation in a person's construction system of religiosity through the processes or phases of anticipation, investment, encounter, confirmation or disconfirmation, and constructive revision; (8) Modulation: Variation in the process of religiosity of an individual, restricted by the permeability of the constructs that serve as guides for new events; (9) Fragmentation: Employment of a variety of construction subsystems for analyzing religiosity; (10) Commonality: Employment of a construction of religious experience—it has similar processes to those of other individuals; and (11) Sociality: Construing the construction processes of another may involve social processes of others related to religiosity.

The CRS has five dimensions. The first dimension is the intellectual dimension, which is the expectation of society that religious people can master some religious knowledge and express their opinions about transcendence, religion, and religiosity. One of the indicators is that religious people have a relatively high frequency of thinking about and updating religious content and issues. The second dimension is the ideology entailing the social expectation that religious people appear to have plausible views on the actuality and substance of a transcendent truth (e.g., God) and the relationship and connection between transcendence and humanity. The third dimension is the public practice, which includes the expectation of society that religious people affiliate with religious groups and have involvement in religious rituals and communal events. This is partly reflected by people's actions and frequency of participation in religious activities, such as attendance at churches for Christians or Muslims prayers on Fridays.

Private practice is the fourth dimension, which concerns the expectation of society that religious people would engage in dialogical and participative patterns of spirituality, which refer to the "one-to-one experience" and "experiences of being at one", respectively. Religious people devote their time to religious activities, such as worship and meditation, in their private space. Religious experience is the last and fifth dimension, which covers the social expectation that religious people (Stark and

Glock 1968, p. 9) encounter emotionality and "some kind of direct contact to ultimate reality" (Huber and Huber 2012, pp. 714–15).

It is notable that the CRS primarily has three versions: 15 items, 10 items, and five items, and they have been applied in different religious traditions with the concept of God, such as Judaism, Christianity, and Islam. To accommodate some of the Eastern religious practices and certain new Western types necessitating meditation as a participative pattern of spirituality, a more general term of "God or something divine" is substituted for the concept of God, and extended to "God, deities, or something divine" for Buddhism and Hinduism (Huber and Huber 2012, p. 719).

*1.2. Related Research in Different Contexts and Sample Representatives*

There was a validation study of the CRS in the Urdu language in Pakistan involving 300 participants from different regions (Abbasi et al. 2019, p. 320). The confirmatory analysis results showed that the CRS in Pakistani culture and Urdu language had three dimensions (renamed as exclusive beliefs, inclusive beliefs, and collective beliefs) and 11 items. Statistical differences were identified in the level of religiosity for shrine believers and non-shrine believers (p. 323). Gheorghe (2019) developed a version for Romania based on the CRS-15 involving more than 200 participants with different religious confessions and an age range of 14 to 51 years in different regions of Romania. The Romanian version reflected good discriminant validity and a high level of reliability, except for item 7 related to the belief in an afterlife (e.g., immortality of the soul, reincarnation), which might give rise to different interpretations from non-Christian religious traditions (Gheorghe 2019, p. 8 of 9).

A study in Poland investigated the relationship between the centrality of religiosity and the coherence sensation for young, median, and older people (Zarzycka 2009; Zarzycka and Rydz 2014). In the study, the Cronbach's alpha for the CRS Polish version was 0.94 with coefficients of the sub-scales ranging from 0.80 to 0.89 (Zarzycka and Rydz 2014, p. 129). In another study in Poland, the centrality of religiosity was found to be positively associated with emotion-related coping, avoidance-oriented coping, social diversion, and positive religious coping (Pargament et al. 2000; Krok 2015, p. 26). In Greece, there was a study analyzing the Greek-translated version of CRS (15 items) in the field of nursing involving more than 300 nurses (Fradelos et al. 2018, p. 26). The results revealed that two factors for five items could be identified: Religious practices and religious beliefs and experiences (p. 28).

The CRS has been used in the African context. Asamani and Mensah (2016, p. 41) used the CRS to investigate religiosity as an antecedent of employees' organizational citizenship behavior in the central region of Ghana. The reliability coefficient of the CRS in that study was 0.827. Esperandio et al. (2019) developed a Brazilian version of the CRS and the validation analyses revealed that a five-factor solution of CRS-10 with 10 items (CRS-10BR) demonstrated better-fit indexes than the single-factor solution of five items (CRS-5BR).

In Asia, a study in Cebu City, Philippines was conducted to probe into the relationship between religiosity as a predictor and the willingness to help an out-group (Batara 2018). In that study, total religiosity of CRS-15 resulted in very good reliability at 0.902. It also reflected that the CRS-15 has been linked to religious identity and the importance of faith in everyday life (Huber and Krech 2009; Batara 2018, p. 75).

*1.3. Versions of CRS*

There are six versions of CRS according to different test lengths (for more details, see Appendices B and C). The CRSi-20 is the longest version that contains five dimensions of 20 items and contains five additional items for the interreligious versions in comparison to CRS-15. Those five items are "04b—How often do you meditate?"; "05b—How often do you experience situations in which you have the feeling that you are at one with all?"; "09b—How important is meditation for you?"; "10b—How often do you experience situations in which you have the feeling that you are touched by a divine power?"; "14b—How often do you try to connect to the divine spontaneously when inspired by daily

situations?". The CRS-15 has three items per dimension and deletes those five additional items for the interreligious aspects. The CRS-10 has 10 items by keeping the first two items of each dimension. The CRSi-14 added four additional Items for the interreligious aspects (04b, 05b, 09b, 10b) to CRS-10. The CRS-5 has five items by keeping the first item of each dimension. Based on the CRS-5, the CRSi-7 adds two items (04b, 05b) to the CRS-5.

It is noteworthy that the dimensions of private practice and experience in the "i-versions" are partly operationalized by alternative indicators, e.g., private practice with prayer and meditation. The participants have to answer both questions. However, only one answer is counted in the calculation of the score of the CRS, which indicated a higher frequency. The reason for this procedure is that the CRS is primarily designed as a measure of the general strength of religiosity, which should be as independent as possible from certain religious orientations.

### 1.4. Context and Methodology of Instrument Validation in This Study

Höllinger and Eder (2016) pointed out that "the overwhelming majority of sociological studies on religion focus on the usual Western understanding of the concept" (p. 4) and the meaning of "being religious" fundamentally differs between Western and Eastern Asian societies (p. 6). The CRS was initially developed for the Western context and was extended to Eastern religions by using more general terms. Studies have demonstrated the validation of the factor structure of the CRS in Western culture and in some of Asian countries, such as Cebu (Batara 2018), Thailand, South Korea, and Indonesia (Huber and Huber 2012). As it is one of the biggest Asian countries, whether CRS could be generalized to the Chinese context is worth being explored. Therefore, this study aims to test the validation of the instrument by using samples from Hong Kong, which is a special administrative region of China and is an ethnically diverse city.

This study is probably the first attempt to use the CRS in the context of Hong Kong as part of the People's Republic of China focusing on a specific target group of teachers in primary and secondary schools.

Convenience sampling is adopted for the present study. A total of 671 in-service teachers (male = 171 and female = 499) with different ranges of age and teaching experiences were solicited from 23 primary and secondary schools in Hong Kong. The respondents have a range of religious backgrounds with Christianity (n = 213), Catholic (n = 22), Buddhism (n = 27), Daoism (n = 3), and no religion (n = 379). It is notable that Hong Kong, formerly a British colony, is a multicultural international city with a wide array of religions. There is also a range of religious and charitable school sponsoring bodies (SSBs) that run and manage schools with governmental support of funding. Some of these major religious SSBs include the Hong Kong Council of the Church of Christ in China, Catholic Diocese of Hong Kong, Caritas Hong Kong, Hong Kong Sheng Kung Hui, Methodist Church Hong Kong, The Evangelical Lutheran Church of Hong Kong, Kowloon Tong Church of the Chinese Christian and Missionary Alliance, Hong Kong Buddhist Association, Hong Kong Taoist Association, etc.

The full English version (Huber and Huber 2012), with permission from Professor Stefan Huber, was translated into Chinese (please refer to Appendix B). A Chinese speaker majoring in English was invited to do the forward translation, and then backward translation was done by a senior research assistant. Two academic colleagues were then invited to help check the translation and help finalize the translated Chinese version.

## 2. Method

### 2.1. Participants

Participants were 671 teachers from Hong Kong. A total of 28.9% of the participants are teachers for students of primary 1–3, 36.4% of the participants are teachers for students from primary 4–6, 13.1% of them are teachers for levels 1–3 of secondary school, and 21% of them are teachers for level 4 or above of secondary school, with 0.6% missing (n = 4).

A total of 23.3% of the participants are below 30 years old (n = 156), 60.9% of the participants ranged in age from 31 to 50 (n = 409), and 14.2% (n = 95) of the participants ranged in age from 51 to 60, with 1.6% missing (n = 11). The sample was 24.7% male (n = 166) and 33.9% female (n = 496), with 1.6% missing data (n = 11).

A total of 24.1% of the participants have 5 years or less teaching experiences, 10.6% of the participants have 6–10 years teaching experiences, 17% of the participants have 11–15 years of teaching experiences, and 48.2% of the participants have 16 years or above teaching experiences, with 0.1% missing data (n = 1).

A total of 36% of the participants are Christian (n = 235), 4.1% are Buddhist (n = 27), 0.5% are Daoist, 1.4% of them are others (n = 9), and 58% of them are nonreligious people (n = 379).

A total of 36.2% of the participants hold master degrees (n = 243), 17% of the participants hold post-bachelor degrees (n = 114), 42.9% of the participants hold bachelor degrees (n = 288), 2.5% of the participants have Postgraduate Diploma in Education (PGDE) diplomas (n = 17), and 1% participants have other diploma (n = 7), with 0.1% missing (n = 1).

## 2.2. Recoding

Three items pertaining to prayer, meditation, and religious services were recoded. For prayer ("How often do you pray?") and meditation ("How often do you meditate"), the original response categories were 1—"Several times a day, 2—"Once a day", 3—"More than once a week", 4—"Once a week", 5—"One or three times a month", 6—"A few times a year", 7—"Less often", and 8—"Never", which were reversed into five categories. Response 1 and 2 was recoded into 5, 3 into 4, 4 and 5 into 3, 6 and 7 into 2, and 8 into 1.

For the question related to religious services ("How often do you take part in religious services?"), the original response categories were 1—"More than once a week", 2—"Once a week", 3—"One or three times a month", 4—"A few times a year", 5—"Less often", and 6—"Never". Responses 1 and 2 were recoded into 5, 3 into 4, 4 into 3, 5 into 2, and 6 into 1.

## 2.3. Data Analyses

The correlations among the 20 items ranged from 0.35 to 0.84 (Appendix D). Confirmatory factor analysis (CFA) was used to examine the structure of the CRS with an aim of establishing an acceptable model. First, we have calculated unidimensional models for all versions of the CRS, in which all manifest variables are explained by only one latent variable. In a second step, we additionally calculated oblique models with five latent variables that correlate with each other for those versions of the CRS where at least two indicators are available for each of the five dimensions (CRS-10, CRS-15, CRSi-14, CRSi-20). These five latent variables correspond to the five dimensions.

For testing the impact of covariates on the defined factor structure, multiple-indicators multiple-causes (MIMIC) analysis was used. As a special case of structural equation modeling (SEM), two components of the MIMIC model are presented: The measuring model section defined the relationships between the latent variable and its indicators; the latent variables were examined, and the casual effects were tested (Joreskog and Sorbom 1996).

The fit of CFA and MIMIC was measured using the Comparative Fit Index (CFI), the Tucker–Lewis Index (TLI), and the Root Mean Square Error of Approximation (RMSEA) method (Schermelleh-Engel et al. 2003). Values of the TLI and CFI greater than 0.95 or RMSEA equal to or smaller than 0.06 indicated a relatively good model fit (Hu and Bentler 1999). Hu and Bentler (1999) also suggested using the combinational rules of those indexes to determine the model fit.

## 3. Results

### 3.1. CFA and Reliability

First, the study tried unidimensional models for all versions of the CRS by treating all the items as one dimension. The relative model fit was good, and the model fit indexes are summarized in Table A1 (Appendix A).

For CRSi-7, the relative model fit indices are relatively good, with CFI = 0.996 and TLI = 0.991, for which RMSEA = 0.096 and Weighted Root Mean Square Residual (WRMR) = 0.710. The factor loadings of all five items were salient and significant, ranging from 0.71 to 0.89. The Cronbach's alpha for CRSi-7 was 0.887. For the CRS-5 version, the relative model fit indices were good, with CFI = 0.993 and TLI = 0.986, while RMSEA = 0.15 and WRMR = 1.017. The factor loadings of all of the five items were salient, ranging from 0.80 to 0.94. The Cronbach's alpha for the five-item version of the CRS was 0.917.

For the CRS-10, the relative model fit indices are relatively good, with CFI = 0.966 and TLI = 0.956, for which RMSEA = 0.209 and WRMR = 2.787. The factor loadings of all 10 items were salient and significant, ranging from 0.71 to 0.94, The Cronbach's alpha for the CRS-10 was 0.957.

For the CRS-15, the relative model fit indices are relatively good, with CFI = 0.964 and TLI = 0.957, for which RMSEA = 0.182 and WRMR = 3.246. The factor loadings of all 15 items were salient and significant, ranging from 0.68 to 0.95. The Cronbach's alpha for the CRS-15 was 0.965.

For the CRSi-14, the relative model fit indices are relatively good, with CFI = 0.975 and TLI = 0.967, for which RMSEA = 0.152 and WRMR = 1.924. The factor loadings of all 14 items were salient and significant, ranging from 0.70 to 0.92. The Cronbach's alpha for the CRSi-14 was 0.942.

For the CRSi-20, the relative model fit indices are relatively good, with CFI = 0.955 and TLI = 0.947, for which RMSEA = 0.171 and WRMR = 2.990. The factor loadings of all 20 items were salient and significant, ranging from 0.70 to 0.93. The Cronbach's alpha for the CRSi-20 was 0.960.

All of these model fit indices satisfied their corresponding cut-off values (0.95) except for the values of RMSEA, which were above 0.1.

The Cronbach's alphas of the five domains for the 20 item version of the CRS—intellectual, ideology, public practices, private practice, and religious experience—are 0.831, 0.837, 0.884, 0.890, and 0.882, respectively. The Cronbach's alphas of the five domains for the 14 item version of the CRS—intellectual, ideology, public practices, private practice, and religious experience—are 0.802, 0.784, 0.764, 0.895, and 0.792, respectively. The Cronbach's alphas of the five domains for the 10 item version of the CRS—intellectual, ideology, public practices, private practice, and religious experience—are 0.802, 0.784, 0.764, 0.920, and 0.910, respectively.

Second, the study calculated oblique models with five latent variables that correlate with each other for those versions of the CRS where at least two indicators are available for each of the five dimensions (CRS-10, CRS-15, CRSi-14, CRSi-20). The results are summarized in Table A2 (Appendix A). For CRSi-20, CRSi-14, and CRS-10, CFA failed, as their latent variable covariance matrixes (psi) were not positive definite. By checking the models, the study found the reason was that the standardized correlations among two of those five subscales of the three versions are greater than one. For CRSi-20, the standardized correlations between intellect and private practice are equal to 1. For CRSi-14 and CRS-10, the standardized correlations between intellect and ideology or public practice are over 1.

For CRS-15, the relative model fit indices were relatively good, with CFI = 0.981 and TLI = 0.976, for which RMSEA = 0.14 and WRMR = 1.95. The factor loadings of all 15 items were salient and significant, ranging from 0.71 to 0.97. The Cronbach's alphas of the five domains for 15 item version of the CRS—intellectual, ideology, public practices, private practice, and religious experience—were 0.831, 0.837, 0.884, 0.930, and 0.937, respectively.

### 3.2. Descriptive Statistics of CRS-5, CRSi-7, and CRS-15

Since the latent variable covariance matrix (PSI) of the other three versions of the CRS (CRS-10, CRSi-14, and CRSi-20) are not positive and the unidimensions of those scales are not often used

in current research, the study only focused on the other three version of CRS—CRS-5, CRSi-7, and CRS-15—in the following analysis.

Table A3 (Appendix A) shows the descriptive statistics of CRS-5, CRSi-7, and CRS-15. Teachers' mean scores of CRS were 2.85, 2.76, and 2.83 for CRS-5, CRSi-7, and CRS-15, respectively. Females scored slightly higher than males on CRS-5, CRSi-7, and CRS-15, while the differences were not statistically significant.

In general, there were statistically significant differences between teachers who had a religious belief and those who did not. Teachers who had religious beliefs scored higher than those who did not have religious beliefs no matter the version.

Based on the cutoff values of categorization of Huber and Huber (2012): Non-religious—1.0 to 2.0; religious—2.1 to 3.9: highly religious—4.0 to 5.0, this study in Hong Kong also created the Norm values of CRS-5, CRSi-7, and CRS-15, as shown in Table A4 (Appendix A).

Based on the results of CRSi-7, the mean (3.00) and standard deviation (SD) (1.04) of CRSi-7 in this study were close to those of Australia (M = 2.97, SD = 1.19) by comparing the Table A5 in the study of Huber and Huber (2012, p. 722). The chi-square test was used to test differences of norm values between this study and Huber and Huber (2012), the results showed that there were statistically significant differences on norm values between Hong Kong and all the other countries provided in Huber and Huber (2012) (see Appendix E).

### 3.3. Comparisons of CRS Score Categorization

Another comparison was made in the study for the CRS-5, CRSi-7, and CRS-15 by categorizing the individuals with reference to the method proposed by Huber and Huber (2012) (see Table A5, Appendix A). The kappa index of agreement was computed, which measures the extent of agreement for categorizing people by a different method. The Kappa for CRS-5 and CRS-15 was the highest (Kappa = 0.865; $p < 0.001$), for CRS-7 and CRS-15 was 0.749 ($p < 0.001$), and for CRS-5 and CRSi-7 was 0.775 ($p < 0.001$). The Friedman test was used to test the differences of categorization by using the three versions; the results showed that $\chi^2 = 110.3$ and $p < 0.001$, that is, there were statistical differences among the three versions when used for categorization.

We also tried another comparison by categorizing individuals in different groups. Table A6a (Appendix A) presented CRS categorizations of religiosity, gender, and educational level. For teachers who have religious beliefs, they showed larger groups of "Highly Religious" individuals, while for those who do not have religious beliefs, there were no teachers categorized into the "Highly Religious" group except for CRSi-7 (0.3%). For CRS-5, CRSi-7, and CRS-15, males have relatively large groups of "Highly Religious" compared to females. All teachers who have a Postgraduate Diploma in Education (PGDE), the professional qualification for teaching, were categorized into the religious group if using CRS-5 and CRS-15. Most of the teachers who have bachelor or post-bachelor degrees were "religious" no matter which version used. For teachers who have a master degree, a larger group of teachers are "religious" when using CRSi-7 and CRS-15. Table A6b (Appendix A) presented CRS categorization between religions. Christians showed a larger ratio in highly religious individuals. More than half of the group of "I have no religion" were categorized as religious in the three versions.

### 3.4. MIMIC Modeling for CRS-5

Figure A1 (Appendix A) displays a MIMIC model constructed for the five-item version of the CRS in this study. The right side is the measurement model, and the left side is the structural model. The model indicated an acceptable relative fit to the data (CFI = 0.971; TLI = 0.959; RMSEA = 0.088).

The analysis showed that gender and religious belief were significant predictors of centrality of religiosity scores after controlling for age and years of teaching experience and education level ($\beta_1 = -0.140$, $p < 0.001$; $\beta_2 = 0.733$, $p < 0.001$). That is, female teachers had higher scores on centrality of religiosity than male teachers. Teachers who had religious beliefs had higher scores on centrality of religiosity than those who do not have religious beliefs.

### 3.5. MIMIC Modeling for CRSi-7

Figure A2 (Appendix A) shows the MIMIC model constructed for the seven-item version of the CRS in this study. The right side is the measurement model, and the left side is the structural model. The model showed acceptable relative fit to the data (CFI = 0.964; TLI = 0.950; RMSEA = 0.084).

The results are similar to those of CRS-5. Gender and religious belief were significant predictors of centrality of religiosity scores after controlling for age and teachers' length of teaching and education level ($\beta_1 = -0.148$, $p < 0.001$; $\beta_2 = 0.698$, $p < 0.001$). That is, female teachers had higher scores on centrality of religiosity than male teachers. Teachers who had religious beliefs had higher scores on centrality of religiosity than those who do not have a religious belief.

### 3.6. MIMIC Modeling for 15 Item Version of the CRS

Figure A3 (Appendix A) shows the MIMIC model constructed for the 15 item version of the CRS in this study. The model showed acceptable relative fit to the data (CFI = 0.955; TLI = 0.938; RMSEA = 0.116).

Table A7 (Appendix A) presents the results of the MIMIC model for CRS-15. The analysis showed that religious belief was a significant positive predictor of all five dimensions of centrality of religiosity scores (intellect, $\beta_1 = 0.636$, $p < 0.001$; ideology, $\beta_2 = 0.602$, $p < 0.001$; public practice, $\beta_3 = 0.634$, $p < 0.001$; private practice, $\beta_4 = 0.673$, $p < 0.001$; experience, $\beta_5 = 0.693$, $p < 0.001$), that is, teachers who had religious beliefs had higher scores on centrality of religiosity than teachers who did not have religious beliefs. Gender was a significant negative predictor of four dimensions (intellect, $\beta_1 = -0.115$, $p < 0.001$; ideology, $\beta_2 = -0.067$, $p < 0.05$; private practice, $\beta_3 = -0.098$, $p < 0.01$; experience, $\beta_4 = -0.157$, $p < 0.001$) of centrality of religiosity scores except for public practice ($\beta = 0.005$, $p > 0.05$), that is, females scored higher on those four dimensions except for public practice. Age was a significant positive predictor for private practice ($\beta = 0.152$, $p < 0.05$), that is, teachers scored higher on private practice with increasing age. Teachers' education level was significantly positively related to private practice ($\beta = 0.070$, $p < 0.05$), that is, teachers who had a higher education level also scored higher on private practice. Teachers' length of teaching experience was not statistically significant in our study.

## 4. Concluding Remarks

The study examined the validation of six versions of the CRS by using samples from the Hong Kong Special Administrative Region (SAR) of the People's Republic of China; the results showed that CRS-15, CRSi-7, and CRS-5 had good or relatively good relative model fit indices. In particular, CRS-15 and CRSi-7, which have been used in many other countries and places, are also valid and reliable measures for centrality of religiosity in the Chinese context in places like Hong Kong, and provide evidence to support the usefulness of the CRS in the Chinese context. All CFI and TLI coefficients are good, as they are above 0.95. The CFI indicates how well the variance of the data is represented by the models. A CFI of 0.96 (rounded up) indicates that 96% of the sample data are represented by the model, which is an acceptable value. However, all RMSEA coefficients are unacceptable because they are greater than 0.100 (RMSEA coefficients are reported to three decimal places). The RMSEA estimates how well the model matrix is adapted to the population matrix. Most of the values are above 0.100—in some cases even significant. One possible explanation might be the special sample (teachers), which differs significantly from the Chinese population. That would be a limitation of the study. Further studies might consider using other samples that are more in line with the population.

It is notable that in Table A2 (Appendix A), the mean scores of the experience dimension or sub-scale had relatively lower mean scores as compared with other dimensions. From a religious perspective, this might suggest more opportunities to be provided for religious and non-religious individuals to engage in two basic forms of solely or dialogically (with one another) in experiencing transcendence.

A contribution of the study is the use of the MIMIC model as the technique to assess the effect of gender and religious beliefs on the Centrality of Religiosity Scale. The conventional techniques, such

as the *t*-test or Analysis of Variance (ANOVA), can only applied to observed variables, while MIMIC could estimate group differences on latent variables by regression on possible cause indicators (Yang et al. 2016). In this study, we found statistical differences in the CRS by gender and religious belief (have or do not have religious belief). No matter which version of the CRS was used, female teachers have higher CRS scores, which was consistent with previous studies, which found that females tend to be more religious than males (Bradshaw and Ellison 2009; Miller and Hoffmann 1995); one of the possible reasons might be that women's beliefs in relating and providing God images are stronger than those of men (Nguyen and Zuckerman 2016), which is worth being studied in the future.

It also makes sense that for people who have religious belonging, they would agree more on the items of the CRS; thus, they would score higher on the CRS.

MIMIC of CRS-15 found that teachers scored higher on private practice with increasing age, which is also consistent with previous studies, which found that individuals tend to become more religious as they age (Stearns et al. 2018; Seifert 2002); as people grow older, they seek religious activities to cope with stress (Idler et al. 2009). The study also found that teachers' education level was significantly positively related to private practice, which was inconsistent with the study of Albrecht and Heaton (1984), who found a negative relationship between education level and religiosity. One of the reasons might be the different contexts of the two studies (Asia vs. Western country); the other reason might be the different scales of religiosity used by the two studies.

Based on the MIMIC results, the study would recommend that future studies should consider gender and participants' religious belonging when using those three versions in the Chinese context. If future readers want to choose the CRS-15, they should also take age and education level into consideration in their study.

There are some limitations to this study. Some other measures related to religiosity and well-being could be used together to provide evidence of predictive and convergent validity. Given the usefulness and validity of the interreligious CRS Chinese version (CRS-5, CRSi-7 and CRS-15), more studies could be extended to other community groups with different age ranges in Hong Kong and the Greater China region to further assess the psychometrics of the CRS.

Qualitative studies could be conducted to unpack the complicated relationships among religiosity, age, teaching qualification, and gender for teachers' variations in their religious experiences.

For the teacher group, studies could be conducted to explore the linkage of the CRS in the Chinese context with their well-being, including happiness (Yorulmaz 2016) and mental health (Hackney and Sanders 2003), as well as possible associations with emotional regulation (Deepika and Gundanna 2018) and citizenship variables (Turska-Kawa 2018).

**Author Contributions:** Conceptualization, J.C.-K.L.; Data curation, X.K.; Formal analysis, X.K.; Methodology, J.C.-K.L.; Supervision, J.C-K.L.; Validation, J.C.-K.L.; Writing–original draft, J.C.-K.L. and X.K. All authors have read and agreed to the published version of the manuscript.

**Funding:** This research received no external funding.

**Conflicts of Interest:** The authors declare no conflict of interest.

## Appendix A

**Table A1.** Confirmatory factor analysis (CFA) results of unidimensional models of all versions of the Centrality of Religiosity Scale (CRS).

|  | CFI | TLI | RMSEA (90% CI) | WRMR | alpha |
|---|---|---|---|---|---|
| CRSi-20 | 0.955 | 0.947 | 0.171 (0.164–0.177) | 2.990 | 0.960 |
| CRS-15 | 0.964 | 0.957 | 0.182 (0.175–0.189) | 3.246 | 0.965 |
| CRSi-14 | 0.975 | 0.967 | 0.152 (0.142–0.164) | 1.924 | 0.942 |
| CRS-10 | 0.966 | 0.956 | 0.209 (0.199–0.220) | 2.787 | 0.957 |
| CRSi-7 | 0.996 | 0.991 | 0.096 (0.068–0.127) | 0.710 | 0.887 |
| CRS-5 | 0.993 | 0.986 | 0.163 (0.135–0.192) | 1.017 | 0.917 |

**Table A2.** CFA results of oblique models of CRS.

| | CFI | TLI | RMSEA (90% CI) | Model Problem |
|---|---|---|---|---|
| CRSi-20 | 0.974 | 0.966 | 0.137 (0.130–0.145) | (PSI) IS NOT POSITIVE<br>Practice and intellect correlation = 1 |
| CRS-15 | 0.981 | 0.976 | 0.138 (0.131–0.145) | II |
| CRSi-14 | 0.984 | 0.971 | 0.144 (0.131–0.157) | (PSI) IS NOT POSITIVE<br>Correlations of several subdimension >1 For |
| CRS-10 | 0.986 | 0.975 | 0.159 (0.146–0.172) | (PSI) IS NOT POSITIVE<br>Correlations of several subdimension >1 |

**Table A3.** Descriptive statistic of CRS-5, CRSi-7, and CRS-15.

| | Total | Gender | | Religious Belief | | |
|---|---|---|---|---|---|---|
| | | Female | Male | Yes | No | Cohen's d |
| CRS-5 | 2.85 (1.12) | 2.86 (1.14) | 2.72 (1.00) | 2.85 ***(1.12) | 2.15 (0.51) | 2.19 |
| CRSi-7 | 3.00 (1.04) | 3.01 (1.05) | 2.89 (0.96) | 3.87 ***(0.97) | 2.39 (0.52) | 1.99 |
| CRS-15 | 2.83 (1.02) | 2.82 (1.00) | 2.79 (1.03) | 3.68 ***(0.92) | 2.21 (0.53) | 2.04 |
| Intellect | 2.89 (0.92) | 2.89 (0.88) | 2.82 (0.99) | 2.89 ***(0.92) | 2.43 (0.61) | 1.46 |
| Ideology | 3.11 (1.06) | 3.09 (1.03) | 3.16 (1.16) | 3.12 ***(1.06) | 2.57 (0.74) | 1.53 |
| Public practice | 2.93 (1.24) | 2.89 (1.25) | 2.95 (1.15) | 2.92 ***(1.24) | 2.22 (0.75) | 1.78 |
| Private practice | 2.83 (1.29) | 2.84 (1.28) | 2.73 (1.28) | 2.82 ***(1.29) | 2.05 (0.66) | 2.01 |
| Experience | 2.40 (1.10) | 2.41 (1.11) | 2.3 (1.01) | 2.39 ***(1.10) | 1.8 (0.70) | 1.67 |

*Note.* ***, $p < 0.001$.

**Table A4.** Norm values of CRS-5, CRSi-7, and CRS-15.

| | CRS-Score | CRS-5 | CRSi-7 | CRS-15 |
|---|---|---|---|---|
| "Non-religious" | 1.00 | 13 | 3 | 10 |
| | 1.20 | 15 | 10 | 11 |
| | 1.40 | 15 | 19 | 19 |
| | 1.60 | 24 | 3 | 9 |
| | 1.80 | 52 | 19 | 37 |
| | 2.00 | 68 | 50 | 73 |
| "Religious" | 2.20 | 136 | 121 | 83 |
| | 2.40 | 35 | 71 | 51 |
| | 2.60 | 24 | 48 | 49 |
| | 2.80 | 25 | 31 | 27 |
| | 3.00 | 27 | 41 | 34 |
| | 3.20 | 28 | 28 | 31 |
| | 3.40 | 18 | 19 | 35 |
| | 3.60 | 8 | 15 | 17 |
| | 3.80 | 23 | 24 | 11 |
| "Highly religious" | 4.00 | 22 | 27 | 15 |
| | 4.20 | 27 | 31 | 30 |
| | 4.40 | 11 | 11 | 62 |
| | 4.60 | 36 | 36 | 33 |
| | 4.80 | 38 | 38 | 0 |
| | 5.00 | 26 | 26 | 15 |
| Mean | | 2.85 | 3.00 | 2.83 |
| SD | | 1.12 | 1.04 | 1.02 |

**Table A5.** Categorization by CRS-5, CRSi-7, and CRS-15.

| Categories | Cutoff | CRS-5 | CRSi-7 | CRS-15 |
|---|---|---|---|---|
| Non-religious | 1.0–2.0 | 25.8% (173) | 15.5% (104) | 23.7% (159) |
| Religious | 2.1–3.9 | 58% (389) | 59.3% (398) | 49.9% (335) |
| Highly religious | 4.0–5.0 | 14.6% (98) | 25.2% (169) | 22.4% (150) |

**Table A6.** (**a**) CRS categorization of religious belief, gender, and educational levels. (**b**) CRS categorization between religions.

| Version | Categories | Religious Belief | | Gender | | Education Level | | | |
|---|---|---|---|---|---|---|---|---|---|
| | | No | Yes | Female | Male | PGDE (n = 17) | Bachelor (n = 288) | Postgraduate (n = 114) | Master (n = 243) |
| CRS-5 | Non-religious | 43.5% | 6.6% | 30.2% | 22.3% | 0 | 34.4% | 20.2% | 26.7% |
| | Religious | 56.5% | 35.8% | 47.0% | 53.6% | 100% | 53.1% | 51.8% | 36.2% |
| | Highly religious | 0 | 57.7% | 22.8% | 24.1% | 0 | 12.5% | 28.1% | 37.0% |
| CRi-7 | Non-religious | 22.4% | 5.5% | 15.5% | 16.3% | 0.0% | 25.0% | 8.8% | 9.1% |
| | Religious | 77.3% | 33.9% | 60.5% | 57.8% | 76.5% | 62.5% | 59.6% | 53.5% |
| | Highly religious | 0.3% | 60.6% | 24.0% | 25.9% | 23.5% | 12.5% | 31.6% | 37.4% |
| CRS-15 | Non-religious | 39.8% | 4.5% | 27.8% | 17.2% | 0 | 31.7% | 18.0% | 22.5% |
| | Religious | 60.2% | 39.7% | 50.0% | 59.5% | 100% | 56.1% | 53.2% | 41.9% |
| | Highly religious | 0 | 55.8% | 22.2% | 23.3% | 0 | 12.2% | 28.8% | 35.6% |

(**a**)

| Version | Categories | Christianity (N = 235) | Buddhism (N = 27) | Taoism (N = 3) | I Have No Religion (N = 379) | Others (N = 9) | Total |
|---|---|---|---|---|---|---|---|
| CRSi-7 | Non-Religious | 1.4% | 22.2% | 0.0% | 22.4% | 66.7% | 15.7% |
| | Religious | 30.9% | 70.4% | 100.0% | 77.3% | 33.3% | 60.5% |
| | Highly Religious | 67.7% | 7.4% | 0.0% | 0.3% | 0.0% | 23.8% |
| CRS-5 | Non-Religious | 1.3% | 33.3% | 0.0% | 43.5% | 66.7% | 28.0% |
| | Religious | 32.3% | 59.3% | 100.0% | 56.5% | 33.3% | 47.8% |
| | Highly Religious | 66.4% | 7.4% | 0.0% | 0.0% | 0.0% | 24.2% |
| CRS-15 | Non-Religious | 1.3% | 11.1% | 0.0% | 39.8% | 66.7% | 24.8% |
| | Religious | 36.0% | 66.7% | 100.0% | 60.2% | 33.3% | 51.4% |
| | Highly Religious | 62.7% | 22.2% | 0.0% | 0.0% | 0.0% | 23.8% |

(**b**)

**Table A7.** Results of the multiple-indicators multiple-causes (MIMIC) model for CRS-15.

| | Intellect | Ideology | Public Practice | Private Practice | Experience |
|---|---|---|---|---|---|
| | B(SE) | B(SE) | B(SE) | B(SE) | B(SE) |
| Gender | −0.115 **(0.037) | −0.067 *(0.034) | 0.005(0.033) | −0.098 *(0.031) | −0.157 ***(0.035) |
| Religious | 0.636 ***(0.031) | 0.602 ***(0.029) | 0.634 ***(0.026) | 0.673 ***(0.023) | 0.693 ***(0.024) |
| Age | −0.002(0.071) | −0.046(0.065) | 0.152 *(0.065) | 0.110(0.064) | −0.029(0.063) |
| Years of teaching experience | −0.040(0.074) | 0.032(0.068) | −0.025(0.063) | −0.070(0.065) | −0.083(0.064) |
| Education level | 0.033(0.040) | 0.032(0.035) | −0.019(0.034) | 0.070*(0.032) | −0.022(0.035) |

Note: *, $p < 0.05$; **, $p < 0.01$; ***, $p < 0.001$.

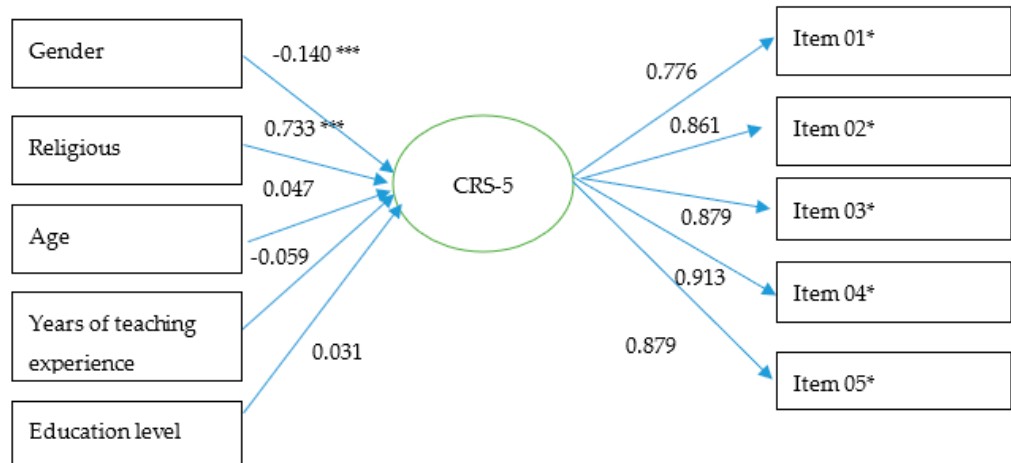

**Figure A1.** MIMIC model for CRS-5; * Item 01 to Item 05 refer to Item 01–05 in Appendix C; *** *p* < 0.001.

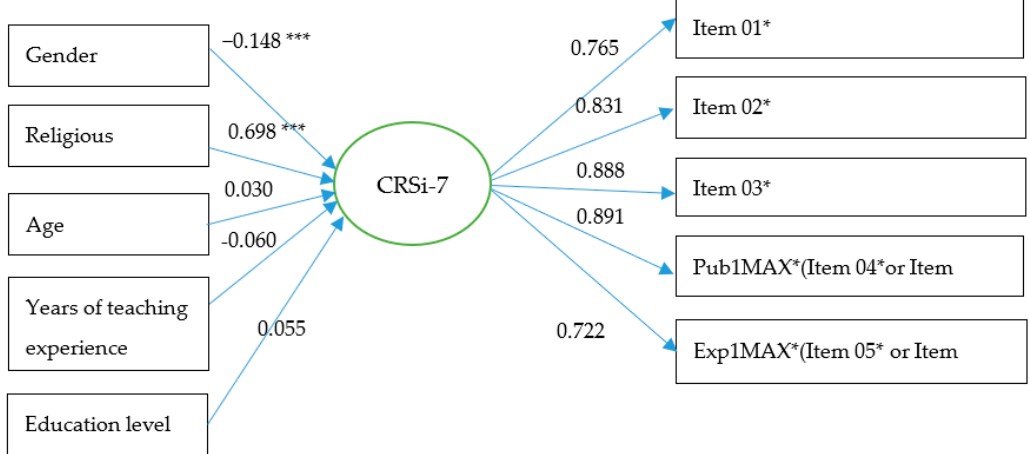

**Figure A2.** MIMIC model for CRSi-7; * Item 01 to Item 05 refer to Item 01-05 in Appendix C; *** *p* < 0.001; Pub1MAX*: The participants answered Item 04(How often do you pray?) and 04b (How often do you meditate?). However, only one answer is counted in the calculation of the score of the CRS, which indicated a higher frequency; Exp1MAX*: The participants answered Item 05(How often do you experience situations in which you have the feeling that God or something divine intervenes in your life?) and 05b (How often do you experience situations in which you have the feeling that you are in one with all?). However, only one answer is counted in the calculation of the score of the CRS, which indicated a higher frequency.

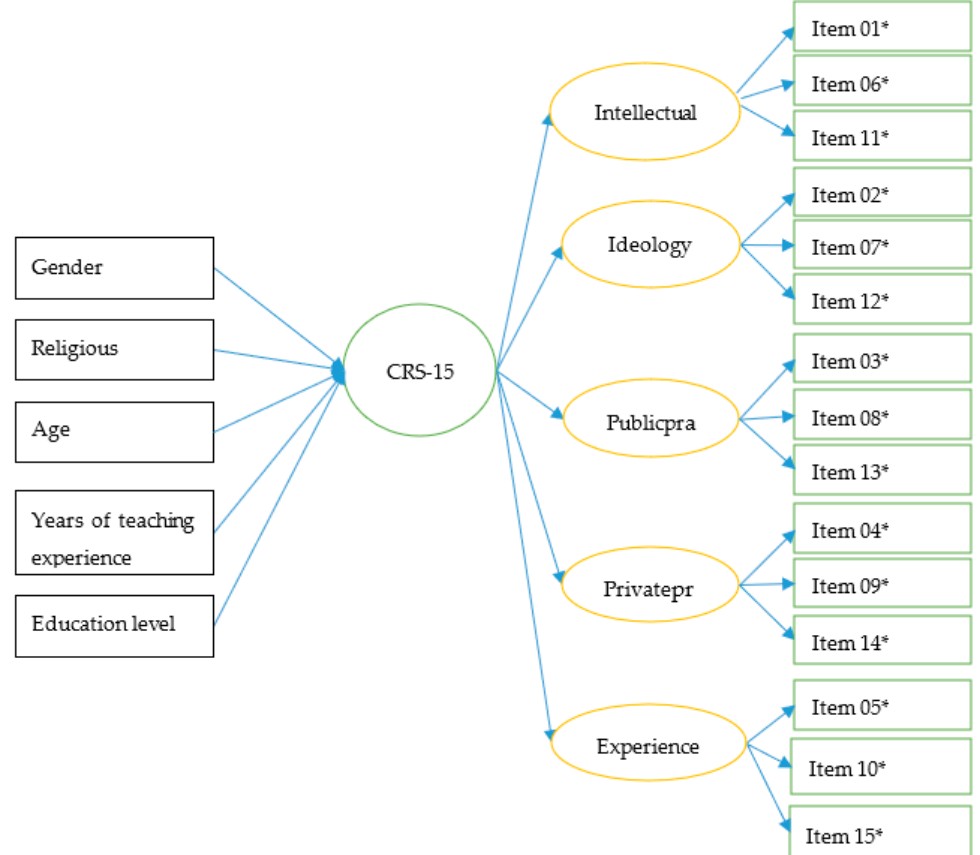

**Figure A3.** MIMIC model for CRS-15; * Items 01–15 refer to the Item 01–15 in Appendix C.

**Appendix B. Attachment: Bilingual version of the Centrality of Religiosity Scale (CRSi-20) in Chinese and English**

請跟據個人對於「神」或「神靈」的觀感來回應以下題目。
**Regarding this one and the following questions, please use your personal imagination of "God" or "something divine".**

1. 在多大程度上相信神或神靈的存在？
   **To what extent do you believe that God or something divine exists?**
   ① 完全不重要 ② 不太重要 ③ 適當 ④ 頗為重要 ⑤ 非常重要
   ① Not at all ② Not very much ③ Moderately ④ Quite a bit ⑤ Very much so
2. 對學習更多的宗教議題有多大興趣？
   **How interested are you in learning more about religious topics?**
   ① 完全不重要 ② 不太重要 ③ 適當 ④ 頗為重要 ⑤ 非常重要
   ① Not at all ② Not very much ③ Moderately ④ Quite a bit ⑤ Very much so
3. 您有多大程度相信來世，例如靈魂的永生、死後復活或輪回轉世之說？
   **To what extent do you believe in an afterlife—e.g., immortality of the soul, resurrection of the dead, or reincarnation?**
   ① 完全不重要 ② 不太重要 ③ 適當 ④ 頗為重要 ⑤ 非常重要
   ① Not at all ② Not very much ③ Moderately ④ Quite a bit ⑤ Very much so
4. 您認為參與宗教儀式有多重要？
   **How important is it to take part in religious services?**
   ① 完全不重要 ② 不太重要 ③ 適當 ④ 頗為重要 ⑤ 非常重要
   ① Not at all ② Not very much ③ Moderately ④ Quite a bit ⑤ Very much so

5. 對您而言，禱告有多重要？
   **How important is a personal prayer for you?**
   ① 完全不重要 ② 不太重要 ③ 適當 ④ 頗為重要 ⑤ 非常重要
   ① Not at all ② Not very much ③ Moderately ④ Quite a bit ⑤ Very much so

6. 對您而言，更高的力量有多大可能存在？
   **In your opinion, how probable is it that a higher power really exists?**
   ① 完全不重要 ② 不太重要 ③ 適當 ④ 頗為重要 ⑤ 非常重要
   ① Not at all ② Not very much ③ Moderately ④ Quite a bit ⑤ Very much so

7. 對您而言，與宗教團體接觸有多重要？
   **How important is it for you to be connected to a religious community?**
   ① 完全不重要 ② 不太重要 ③ 適當 ④ 頗為重要 ⑤ 非常重要
   ① Not at all ② Not very much ③ Moderately ④ Quite a bit ⑤ Very much so

8. 對您而言，冥想有多重要？
   **How important is meditation for you?**
   ① 完全不重要 ② 不太重要 ③ 適當 ④ 頗為重要 ⑤ 非常重要
   ① Not at all ② Not very much ③ Moderately ④ Quite a bit ⑤ Very much so

9. 您多久禱告一次？
   **How often do you pray?**
   ① 每天多次 ② 每天一次 ③ 一星期多於一次 ④ 一星期一次
   ⑤ 一個月幾次 ⑥ 一年幾次 ⑦ 甚少 ⑧ 從不
   ① Several times a day ② Once a day ③ More than once a week ④ Once a week ⑤ One to three times a month ⑥ A few time a year ⑦ Less often ⑧ Never

10. 您多久冥想一次？
    **How often do you meditate?**
    ① 每天多次 ② 每天一次 ③ 一星期多於一次 ④ 一星期一次 ⑤ 一個月幾次 ⑥ 一年幾次 ⑦ 甚少 ⑧ 從不
    ① Several times a day ② Once a day ③ More than once a week ④ Once a week ⑤ One to three times a month ⑥ A few time a year ⑦ Less often ⑧ Never

11. 您多久參與一次宗教儀式？
    **How often do you take part in religious services?**
    ① 每天多次 ② 每天一次 ③ 一星期多於一次 ④ 一星期一次 ⑤ 一個月幾次 ⑥ 一年幾次 ⑦ 甚少 ⑧ 從不
    ① Several times a day ② Once a day ③ More than once a week ④ Once a week ⑤ One to three times a month ⑥ A few time a year ⑦ Less often ⑧ Never

您多常經歷以下的情況或事件？
**How often do you experience the following situations or events?**

12. 您多久思考一次與宗教相關的事情？
    **How often do you think about religious issues?**
    ① 從不 ② 甚少 ③ 有時 ④ 時常 ⑤ 經常
    ① Never ② Rarely ③ Occasionally ④ Often ⑤ Very often

13. 您多久經歷一次神或神靈在影響你的生活？
    **How often do you experience situations in which you have the feeling that God or something divine intervenes in your life?**
    ① 從不 ② 甚少 ③ 有時 ④ 時常 ⑤ 經常
    ① Never ② Rarely ③ Occasionally ④ Often ⑤ Very often

14. 您多久經歷一次神或神靈希望向你展示或啓示某些東西？

**How often do you experience situations in which you have the feeling that God or something divine wants to show or reveal something to you?**

① 從不 ② 甚少 ③ 有時 ④ 時常 ⑤ 經常

① Never ② Rarely ③ Occasionally ④ Often ⑤ Very often

15. 您多久透過媒體（例如電台、電視、網絡、報紙或書籍）接收一次有關宗教的資訊？

**How often do you keep yourself informed about religious questions through radio, television, internet, newspapers, or books?**

① 從不 ② 甚少 ③ 有時 ④ 時常 ⑤ 經常

① Never ② Rarely ③ Occasionally ④ Often ⑤ Very often

16. 當受到日常生活發時，您多久會自發地禱告一次？

**How often do you pray spontaneously when inspired by daily situations?**

① 從不 ② 甚少 ③ 有時 ④ 時常 ⑤ 經常

① Never ② Rarely ③ Occasionally ④ Often ⑤ Very often

17. 您多久經歷一次神或其他聖靈的同在？

**How often do you experience situations in which you have the feeling that God or something divine is present?**

① 從不 ② 甚少 ③ 有時 ④ 時常 ⑤ 經常

① Never ② Rarely ③ Occasionally ④ Often ⑤ Very often

18. 您多久經歷一次自己與神或聖靈「合而為一」的感覺？

**How often do you experience situations in which you have the feeling that you are at one with all?**

① 從不 ② 甚少 ③ 有時 ④ 時常 ⑤ 經常

① Never ② Rarely ③ Occasionally ④ Often ⑤ Very often

19. 您多久經歷一次你被神聖力量觸碰到？

**How often do you experience situations in which you have the feeling that you are touched by a divine power?**

① 從不 ② 甚少 ③ 有時 ④ 時常 ⑤ 經常

① Never ② Rarely ③ Occasionally ④ Often ⑤ Very often

20. 當受到日常生活發時，您多久嘗試與神靈接觸一次？

**How often do you try to connect to the divine spontaneously when inspired by daily situations?**

① 從不 ② 甚少 ③ 有時 ④ 時常 ⑤ 經常

① Never ② Rarely ③ Occasionally ④ Often ⑤ Very often

# Appendix C

**Table A8.** Table for Different Versions of the CRS.

| | Items | CRS-20 | CRS-15 | CRS-14 | CRS-10 | CRS-7 | CRS-5 |
|---|---|---|---|---|---|---|---|
| Intellectual | 01: How often do you think about religious issues? | ✓ | ✓ | ✓ | ✓ | ✓ | ✓ |
| | 06: How interested are you in learning more about religious topics? | ✓ | ✓ | ✓ | ✓ | | |
| | 11: How often do you keep yourself informed about religious questions through radio, television, internet, newspapers, or books? | ✓ | ✓ | | | | |
| Ideology | 02: To what extent do you believe that God or something divine exists? | ✓ | ✓ | ✓ | ✓ | ✓ | ✓ |
| | 07: To what extent do you believe in an afterlife—e.g., immortality of the soul, resurrection of the dead, or reincarnation? | ✓ | ✓ | ✓ | ✓ | | |
| | 12: In your opinion, how probable is it that a higher power really exists? | ✓ | ✓ | | | | |
| Public practice | 03 How often do you take part in religious services? | ✓ | ✓ | ✓ | ✓ | ✓ | ✓ |
| | 08 How important is it to take part in religious services? | ✓ | ✓ | ✓ | ✓ | | |
| | 13: How important is it for you to be connected to a religious community? | ✓ | ✓ | | | | |
| Private practice | 04: How often do you pray? | ✓ | ✓ | ✓ | ✓ | ✓ | ✓ |
| | 09: How important is personal prayer for you? | ✓ | ✓ | ✓ | ✓ | | |
| | 14 How often do you pray spontaneously when inspired by daily situations? | ✓ | ✓ | | | | |
| | 04b How often do you meditate? | ✓ | | ✓ | | ✓ | |
| | 09b How important is meditation for you? | ✓ | | | | | |
| | 14b How often do you try to connect to the divine spontaneously when inspired by daily situations? | ✓ | | | | | |
| Experience | 05: How often do you experience situations in which you have the feeling that God or something divine intervenes in your life? | ✓ | ✓ | ✓ | ✓ | ✓ | ✓ |
| | 10: How often do you experience situations in which you have the feeling that God or something divine wants to communicate or to reveal something to you? | ✓ | ✓ | ✓ | ✓ | | |
| | 15: How often do you experience situations in which you have the feeling that God or something divine is present? | ✓ | ✓ | | | | |
| | 05b: How often do you experience situations in which you have the feeling that you are at one with all? | ✓ | | ✓ | | ✓ | |
| | 10b: How often do you experience situations in which you have the feeling that you are touched by the divine? | ✓ | | ✓ | | | |

## Appendix D

**Table A9.** Correlations between Items of CRSi-20.

| | Int1 | Int2 | Int3 | Ide1 | Ide2 | Ide3 | Pub1 | Pub2 | Pub3 | priv1Max | priv2max | priv3max | Exp1Max | Exp2Max | Exp3 |
|---|---|---|---|---|---|---|---|---|---|---|---|---|---|---|---|
| Int1 | 1 | | | | | | | | | | | | | | |
| Int2 | 0.670 ** | 1 | | | | | | | | | | | | | |
| Int3 | 0.624 ** | 0.570 ** | 1 | | | | | | | | | | | | |
| Ide1 | 0.678 ** | 0.769 ** | 0.541 ** | 1 | | | | | | | | | | | |
| Ide2 | 0.634 ** | 0.640 ** | 0.445 ** | 0.645 ** | 1 | | | | | | | | | | |
| Ide3 | 0.580 ** | 0.552 ** | 0.399 ** | 0.554 ** | 0.697 ** | 1 | | | | | | | | | |
| Pub1 | 0.628 ** | 0.538 ** | 0.487 ** | 0.581 ** | 0.381 ** | 0.349 ** | 1 | | | | | | | | |
| Pub2 | 0.726 ** | 0.691 ** | 0.524 ** | 0.772 ** | 0.579 ** | 0.568 ** | 0.622 ** | 1 | | | | | | | |
| Pub3 | 0.717 ** | 0.684 ** | 0.567 ** | 0.758 ** | 0.551 ** | 0.616 ** | 0.676 ** | 0.869 ** | 1 | | | | | | |
| priv1Max | 0.715 ** | 0.698 ** | 0.655 ** | 0.716 ** | 0.565 ** | 0.558 ** | 0.632 ** | 0.658 ** | 0.685 ** | 1 | | | | | |
| priv2max | 0.690 ** | 0.744 ** | 0.556 ** | 0.718 ** | 0.589 ** | 0.580 ** | 0.553 ** | 0.753 ** | 0.781 ** | 0.820 ** | 1 | | | | |
| priv3max | 0.713 ** | 0.670 ** | 0.612 ** | 0.713 ** | 0.532 ** | 0.508 ** | 0.670 ** | 0.699 ** | 0.731 ** | 0.713 ** | 0.667 ** | 1 | | | |
| Exp1Max | 0.609 ** | 0.535 ** | 0.429 ** | 0.585 ** | 0.444 ** | 0.419 ** | 0.532 ** | 0.519 ** | 0.497 ** | 0.528 ** | 0.501 ** | 0.658 ** | 1 | | |
| Exp2Max | 0.659 ** | 0.573 ** | 0.471 ** | 0.597 ** | 0.452 ** | 0.478 ** | 0.512 ** | 0.608 ** | 0.635 ** | 0.615 ** | 0.665 ** | 0.773 ** | 0.659 ** | 1 | |
| Exp3 | 0.652 ** | 0.622 ** | 0.520 ** | 0.637 ** | 0.453 ** | 0.433 ** | 0.618 ** | 0.642 ** | 0.641 ** | 0.623 ** | 0.659 ** | 0.773 ** | 0.637 ** | 0.843 ** | 1 |

Note: ** Correlation is significant at the 0.01 level (two-tailed).

## Appendix E

**Table A10.** Chi-Square test on the Numbers of Norm Values between Hong Kong and Other Countries.

|  |  | $x^2$ | $p$ |
|---|---|---|---|
| Hong Kong | TOTAL | 1695.529a | $p < 0.001$ |
| Hong Kong | WEST | 1481.744a | $p < 0.001$ |
| Hong Kong | EAST | 1927.777a | $p < 0.001$ |
| Hong Kong | MULI | 1260.128a | $p < 0.001$ |
| Hong Kong | AT | 1652.285a | $p < 0.001$ |
| Hong Kong | CH | 1573.741a | $p < 0.001$ |
| Hong Kong | IT | 1251.146a | $p < 0.001$ |
| Hong Kong | FR | 1962.384a | $p < 0.001$ |
| Hong Kong | ES | 1372.525a | $p < 0.001$ |
| Hong Kong | GB | 1653.915a | $p < 0.001$ |
| Hong Kong | PL | 1366.833a | $p < 0.001$ |
| Hong Kong | RU | 1781.262a | $p < 0.001$ |
| Hong Kong | IL | 1302.300a | $p < 0.001$ |
| Hong Kong | TR | 1203.176a | $p < 0.001$ |
| Hong Kong | MA | 1183.734a | $p < 0.001$ |
| Hong Kong | NG | 899.919a | $p < 0.001$ |
| Hong Kong | ID | 1115.860a | $p < 0.001$ |
| Hong Kong | IN | 1139.979a | $p < 0.001$ |
| Hong Kong | TH | 1412.524a | $p < 0.001$ |
| Hong Kong | KR | 1635.977a | $p < 0.001$ |
| Hong Kong | AU | 1593.924a | $p < 0.001$ |
| Hong Kong | US | 1152.901a | $p < 0.001$ |
| Hong Kong | GTGT | 1056.416a | $p < 0.001$ |
| Hong Kong | BRBR | 1000.678a | $p < 0.001$ |

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
