# Peer review of "Validation of the Chinese Version of the Centrality of Religiosity Scale (CRS): Teacher Perspectives"

_religions, doi:10.3390/rel11050266_

Round 1

Reviewer 1 Report

The main issue with this paper was not so much that the case or the data was not interesting or of value. Indeed this is a valuable contribution to the field and the fieldwork is adequately conducted. However, the presentation is a bit "data heavy." The authors should read through the material and increase the amount of narrative and descriptive material about the setting, while reducing the number of charts and diagrams. The amount of raw data presented in the text could then be moved to additional charts in the appendices, further increasing space in the text to focus on narrativity and implications of the study. 

Author Response

Thanks, we have moved all the tables and figures of the text to the Appendix A.

Reviewer 2 Report

I have two principal comments on this otherwise good article:

Line 163 makes a distinction between Christians and Catholics that is not correct. Perhaps this should be a distinction between Protestant Christianity (n = 213), Catholic Christianity (n-22)? Catholics are Christians.

Line 221 does not make sense as an English sentence. Perhaps there are missing words?

Apart from other English editing concerns, the remainder of the article is clear. The research design is appropriate, and results are reported in ways that make the significance of the research clear. The recommendations regarding future directions for additional research are also appropriate.

Author Response

Line163:  We combined them suggested by the reviewers. The following are the original comment:

“The division of participants into the categories of “Christian” and “Catholic” on page 4 line 171 is confusing because Catholic believers are a part of the Christian tradition. It would be more accurate to label these two groups as “Protestant” and “Catholic” or as “Protestant Christian” and “Catholic Christian.” That change should be done consistently throughout the essay, including the discussions on pages 7 and 8.”

Line221:  Thanks, we have changed that into “The value of TLI and CFI greater than 0.95 or RMSEA equal or smaller 0.06 indicated a relatively good model fit (Hu & Bentler, 1999).”

Reviewer 3 Report

line 255: are 0.80 instead are0.802

line 262: point after ,,than one”

line 275: Table 3 instead Table3

line 282: not-religious 1.0 instead not-religious1.0

Author Response

Thanks for your comment, we have made all the changes according to your suggestions.

This manuscript is a resubmission of an earlier submission. The following is a list of the peer review reports and author responses from that submission.

Round 1

Reviewer 1 Report

This essay extends the investigation of the CRS scale to a new context and a new population in helpful and significant ways. It contributes to a wider body of emerging scholarship on the CRS scale. The authors adequately explain their analysis of the different versions of the CRS and the design of their research and statistical analyses.

I have five areas for recommended changes.

The essay needs revision of English to increase clarity and to correct grammar and syntax. I will not comment comprehensively, but here are some examples:

Page 1 line 16 uses “Besides” when you need to use “In addition.”

Page 1 beginning line 35. Revise by breaking apart this long sentence and making the phrases and clauses more parallel. For example, you could write,

There are specific measures of religiosity for some religious groups. Aziz & Rehman (1996), for example, studied Muslims’ constructs of religious effect, doctrine, and faith, while Wilkes, Burnett and Howell (1986) measured Christians’ dimensions of self-perceived religiousness, the significance of religious ethics, church attendance, and assurance in religious norms or confidence in religious values (Khan, 2104, pp. 69-70).

Page 3 line 105 “adults young, moderate and late” should be “young, median, and older” adults.

Page 4 line 164 should read “of the participants ranged in age” and so also on line 165. The language used to discuss missing data is not consistent. The clearest phrasing would be “The sample was ….., with 1.6% missing data.”

The discussion of the versions of the CRS is split into two parts (page 2 lines 87-92 and page 3 lines 125-137). These sections should be combined. I recommend moving the subheading to page 2 line 87 and then consolidating the information before the section on “Related research in different context and sample representatives.”

The division of participants into the categories of “Christian” and “Catholic” on page 4 line 171 is confusing because Catholic believers are a part of the Christian tradition. It would be more accurate to label these two groups as “Protestant” and “Catholic” or as “Protestant Christian” and “Catholic Christian.” That change should be done consistently throughout the essay, including the discussions on pages 7 and 8.

The header for Table 1 on page 5 is not in the right place.

Figure 3 for CRS-15 does not match figures 1 and 2 for the other versions of CRS in its discussion of the MIMIC modelling. There is no explanation for the difference, which is confusing. Either made the three figures consistent or explain why figure 3 is different in terms of the categories it uses. Also, showing the complete questions in the text boxes would be helpful. Some boxes have complete questions, but others are truncated.

Author Response

1) In the correlation matrix of the CRSi-20 only the variable names are listed.

Response: The correlation matrix was redone please check in the revision (Appendix B)

2) I wonder why the models to the CRS-5, CRS-15 and CRSi-7 work and the other three models do not. Can you explain this? Could you send more from the output of MPlus you got? Normally output like the following occurs naming the problematic variables:

Response: Yes, all those other three models have warnings like this captured picture, as I took a look at the output, the standardized correlations between subdimensions equals or are above 1.

Take 14-item as example:

For 20-item :

For 10-item:

3) You have calculated oblique models. As an alternative, have you also calculated one-dimensional models (as for CRS-5 and CRSi-7) for the CRS-10, CRSi-14 and CRSi-20? A third possibility would be to calculate models with secondary factors instead of oblique models. It would be nice if you would try the one-dimensional and secondary factor models and send me the results.

Response: Done~

There are similar problems for second order model;

for 20-item

For 14-item,the results are similar:

For 10-item,the results are similar:

4) A few remarks about the MIMIC models:

- The model for CRSi-7, figure 3, is still calculated with seven instead of five manifest variables. Please use priv1Max and Exp1Max here.

Response:  I have changed that.

5)- The model for CRS-15, figure 3, is still missing the manifest variables and the regression weights.

Response: The regression weights are listed in Table 6 as there are five weighs for each sub dimension. It would be very confusing to put all of them in the diagram. I added manifest variables in the figure.

6) I've also made little correction at the title and in the abstract.

Response: Thanks, we used your version

Reviewer 2 Report

This study has the potential to be published after some revisions, especially regarding its context, purpose and theoretical and practical significance in relation to international studies on religion, belief and religiosity.

The object and context of the study: This study applies the use of Centrality of Religiosity (CRS) to Chinese primary and secondary teachers in Hong Kong in order to validate the instrument for generalization internationally. The study describes other international contexts for instrument validations in the background (p. 3, lines 138-157). Instrument validation should also consider potential critiques of the instrument and alternatives to it. Literature review should be expanded to include alternative contemporary frameworks for the notion of religiosity. Reference to the disciplinary context of religious education states that “religious education tends to highlight orthodoxies and beliefs” (Groome, 1998, p. 1, lines 28-29). This is highly questionable generalization based on one fairly old source, when there is vast body of research literature available.

Participants and methods: Participants were 671 teachers from Hong Kong primary and secondary schools. Background information was described in terms of age, sex, teaching experience and religious affiliation and academic education at degree level. Data collection should be explained in more detail: why is the convenience sampling of teachers (p. 3, line 142) justified in this study? Why are Catholics differentiated from other Christians, but no other Christian denominations are mentioned?

Relevance of study and significance of results: The originality of this study is based on its application to new context and participants, but the purpose and significance of the study is not sufficiently explained: why is it important to validate this intstrument with Chinese teachers in primary and secondary schools? The study addresses its limitations and suggests further studies, (p.2, 356-366). However, I recommend evaluation of the study as whole in relation to its task of validating the instrument. The study needs to clarify and elaborate the following questions: What is the purpose and relevance of conducting a validation of this instrument in the Chinese context? Why are teachers selected as participants in the study? Studying teachers implies potential educational significance, which is not addressed. What is the significance of studying teachers’ religiosity? How does it relate to teaching or the educational context? Who benefits from this study? How does it impact on the Chinese context? What is the disciplinary and societal impact of this study? How do the results compare to and differ from earlier studies on religiosity? I recommend more elaborate discussion on the results in the light of contemporary international and interdisciplinary research literature.

Author Response

(The authors gave the same response as above.)
